REGISTERED REPORT PROTOCOL

# Progressive active mobilization with dose control and training load in critically ill patients (PROMOB): Protocol for a randomized controlled trial

**Rodrigo Santos de Queiroz**[1]◉*, **Micheli Bernardone Saquetto**[2]◉, **Bruno Prata Martinez**[2]◉, **Bianca Bigogno Reis Cazeta**[1]◉, **Carol Hodgson**[3]◉, **Mansueto Gomes-Neto**[2]◉

1 Program in Medicine and Health of the Faculty of Medicine, Federal University of Bahia, Salvador, Brazil,
2 Department of Physical Therapy, Institute of Health Sciences, Federal University of Bahia, Salvador, Brazil,
3 Australian and New Zealand Intensive Care Research Centre, Monash University, Melbourne, Australia

◉ These authors contributed equally to this work.
* rsqueiroz@uesb.edu.br

This is a Registered Report and may have an associated publication; please check the article page on the journal site for any related articles.

## Abstract

The dose of progressive active mobilization is still uncertain. The purpose of this study is to identify if the addition of a protocol of progressive active mobilization with dose and training load control to usual care is effective in reducing the length of stay in intensive care unit (ICU) and the improvement of the functioning, incidence of ICU-acquired weakness (ICUAW), mechanical ventilation duration and mortality rate in patients hospitalized in ICU. It is Double-blind randomised clinical trial. The setting for this trial will be medical and surgical ICU of a university hospital. The study participants will be 118 patients aged> 18 years admitted to ICU for less than 72 hours. Participants will be randomized to either an experimental or control group. The experimental group will undertake addition of a protocol of progressive active mobilization with dose and training load control to usual care, while the control group will undertake only usual care. The primary outcome will be length of ICU stay. The secondary outcomes will be Cross-sectional area and muscle thickness of the rectus femoris and biceps brachii, Change in muscle strength from the baseline, Functional Status, incidence of ICUAW, Days with mechanical ventilation and Mortality. All statistical analyses will be conducted following intention-to-treat principles. It has a detailed description of the dose of exercise, was designed with the strictest methodological criteria. These characteristics allow to investigate with greater certainty the results progressive active mobilization in critical patients, allowing replication and future combinations in meta-analyzes.

## Introduction

Physical rehabilitation strategies involving active and progressive mobilization protocols have been recommended for the management of critical illness-related morbidity [1]. The strategies

**Data Availability Statement:** All relevant data from this study will be made available upon study completion.

**Funding:** The study received funding support from the Brazilian National Research Council (CNPQ) and the Coordenação de Aperfeiçoamento de Pessoal de Nível Superior – Brasil (CAPES) – Finance Code 001. http://www.cnpq.br/ https://www.capes.gov.br/ B.B.R.C received a scholarship NO. The funders had and will not have a role in study design, data collection and analysis, decision to publish, or preparation of the manuscript.

**Competing interests:** The authors have declared that no competing interests exist.

focus on reducing the length of stay in an intensive care unit (ICU LOS), preventing prolonged mechanical ventilation [2], increasing mobility and muscle strength, and extending the days alive out of hospital to 180 days [3] in patients hospitalized in an intensive care unit.

Despite the recommendation of early mobilization for critically ill patients, the effects of mobilization are inconclusive. Serious risk of bias, imprecision, and heterogeneity of individual study designs and intervention protocols hamper the ability to achieve a precise and accurate estimate of the effect size and decrease the quality of evidence [4]. Few published randomized clinical trials (RCTs) have adequately described the dose and intensity of the protocols used [4,5]. There is considerable variability in the results of clinical trials evaluating early mobilization [6]. Tipping et al. [3] reported that there is no consistent effect of active mobilization on ICU LOS. According to the authors [3], several studies did not separate the LOS for survivors and non-survivors and the limitations identified in the individual RCTs prevent the pooling of results for meta-analysis.

It is well documented that the exercise dose is a determining factor for the results of exercise programs. Variables such as intensity, execution time and rest, number of repetitions, and progression may influence strength gain and cardiorespiratory capacity [7,8]. Despite the variety of progressive active mobilization protocols, the optimal training scheme has not yet been defined [3,4]. The most important factor influencing outcomes such as LOS, strength, and function may be the addition of a training dose control protocol based on the major international protocols [5] and involving variables of intensity, frequency, number of repetitions, rest time, and progression scheme.

Therefore, the primary endpoint of this RCT will be to identify if the addition of a protocol of progressive active mobilization with dose and training load control to usual care is effective in reducing the length of stay in ICU. The secondary outcomes will be cross-sectional area and muscle thickness of the rectus femoris and biceps brachii, change in muscle strength, functional status, incidence of ICU-acquiredweakness (ICUAW), days with mechanical ventilation, and mortality.

## Methods

The study will use the single-center, randomized controlled trial protocol, with blinding of the assessors and participants, to evaluate progressive active mobilization compared to usual care in the ICU. This is a study of superiority that adhered to the Consolidated Standards of Reporting Trials (CONSORT) recommendations [9] and to Standard Protocol Items for Clinical Trial (SPIRIT) [10]. Clinical Trial Registration Number: NCT03596853.

### Participants

Patients admitted to the ICU of the Universitary Hospital Professor Edgard Santos of the Federal University of Bahia (UFBA), Salvador, Bahia, Brazil, who met the eligibility criteria will be recruited in the study. Inclusion criteria: Patients hospitalized in the ICU aged 18 years or more with the following: a minimum functional classification of level 1 (i.e. being able to roll in the bed and bridge); Barthel index [11] recall of at least 70, 2 weeks prior to ICU admission [12]; and the ability to interact with the physiotherapist, determined by at least three positive responses of the following five commands: (1) "Open (close) your eyes," (2) "Look at me," (3) "Open your mouth and put out your tongue," (4) "Nod your head" and (5) "Raise your eyebrows when I have counted to five" [13].

Exclusion criteria included the following: patients with hospitalization time greater than 72 hours before the start of the study; patients with an estimated mortality rate greater than 50% according to Acute Physiology and Chronic Health Disease Classification System II (APACHE

II) [12]; patients with increased intracranial pressure or with an history of cardiorespiratory arrest; patients with unstable fractures that hamper the progression of mobilization levels; lower limb amputees; patients with degenerative neuromuscular disease; and those with a history of cerebrovascular accident, traumatic brain injury, or radiotherapy and/or chemotherapy in the last 6 months.

## Randomization and allocation

After the start of the research, all the patients admitted to the ICU will be assessed for inclusion based on the eligibility criteria through daily visits in direct communication with the multidisciplinary team. The patients that met the eligibility criteria will be allocated randomly in the intervention and control groups using the simple randomization strategy [14–16]. The numbers associated with the identities of the eligible patients will be concealed in an opaque, coded, and unidentified envelope [17], which will be signed and dated by the researcher responsible for the allocation. Only the physiotherapist who performed the intervention will know the meaning of the code contained in the envelope and, therefore, will know which patients received the progressive active mobilization protocol. Randomization will be performed by an independent researcher not involved in the study (the independent researcher will not be part of the intervention or the evaluation and will be blinded to the intervention and control codes).

## Blinding

Patients, ultrasound assessors, and functional outcome evaluators will be blind to allocation of patients in the intervention and control groups.

## Assignment of interventions

The control, the usual care group (UCG), will receive mobilization according to the routine of their service: passive kinesiotherapy (consisting of passive range of motion exercises)for patients with a low level of consciousness who are unable to cooperate and perform voluntary active movements; active exercises and mobility training for cooperative patients capable of performing active movements voluntarily; and eventually, use of electrostimulation and passive orthostatism. There is still no specific institutional protocol for mobilization [18]. The UCG mobilization process will be managed by the hospital's team of physiotherapists and without the influence of researchers. The exercises are individualized, applied at low intensity and according to the patient's tolerance, lasting between 10 and 20 minutes, usually twice a day, 7 days a week.

The patients in the intervention group (IG) will undergo a progressive active mobilization protocol with individual dose control and training load stratified according to functional levels and performance (Fig 1) by appropriately trained physiotherapists. The protocol will be applied by two research physiotherapists who have undergone six months of training and who are not part of the hospital's physiotherapist team. Beyond the progressive active mobilization protocol, the IG will also receive the usual care according to the routine of service. Usual care will be provided as described in the control group.

The progressive active mobilization protocol will be applied once a day, five times a week. If the patient performed properly within their functional level but was not able to progress to the next level, then there will be a training volume increase within the level (Fig 1). Performed properly means the patient will be able to complete more than eight series [7] of each exercise and did not present increased sensation of pain [19,20], sensation of perceived exertion within

**Fig 1. Progressive active mobilization protocol stratified by functional levels and performance.**
FSS-ICU = Functional Status Score for the ICU [22]. If need minimal or moderate assistance (FSS-ICU $\geq$ 3) progresses at functional levels ($N_1$, $N_2$, $N_3$, $N_4$); Adequate performance = Able to perform 8 series of each movement, alternating 20 seconds of execution with 10 seconds of rest, without discomfort (Borg < 6; scale 0–10) [20], without increased pain and positive response to the questioning: "*Do you feel good after you do physical activity*?" [21]. If it does not have FSS-ICU $\geq$ 3, but has adequate performance, maintains functional level and increases training volume. The protocol is applied 5 times a week, once a day added to the usual care. If the patient reports discomfort (Borg > 6), increased pain, will rest for 5 minutes. If there is no normalization of the pain and sensation of exertion perceived in that period of time, the training session will be suspended and performed after 24 hours.

the safety limit of the protocol (Borg $\leq$ 6; scale 0–10) [19,21] and positive response to the questioning: "Do you feel good after you do physical activity [19,22].

The training will be performed through progressive functional movements: N1 (bridge and rolling for both sides); N2 (transfer from supine to sit on both sides); N3 (transfer from sit to stand); and N4 (walking). According to their functional level, patients will perform, once a day (under direct supervision of the researchers), eight series of each movement, alternating 20 seconds of execution with 10 seconds of rest [7]. Patients will be stimulated to perform the movements with the highest possible speed and by maintaining a maximum Borg of 6 [21].

Patients will be able to progress even if they were not able to perform the movement independently; however, they should have only needed minimal or moderate assistance from one physiotherapist, reflecting a Functional Status Score for the ICU (FSS-ICU) $\geq$ 3 [23] for a specific activity, like rolling for example. Patients will be able to use a chair with armrests for the sit-and-lift workout and auxiliary devices (crutch, walker and/or cane) for training. If the patient report discomfort (Borg > 6) [21] or increased pain, the patient will be advised to in rest for to 5 minutes. If there is no normalization of pain and perceived effort within 5 minutes, the training session will be suspended and performed again after 24 hours.

After the patient meets the performance criteria to increase the training volume, new exercises will be added following the same criteria of eight series of each movement, alternating 20 seconds of execution with 10 seconds of rest: $N_1$ (maximal triple flexion/extension of lower limbs: involving hip, knee and ankle -consists of flexing and extending the lower limbs alternately as far as possible); $N_2$ (exercise of trunk rotation and anterior and posterior reach of a ball); $N_3$ (sit and stand from a chair throwing a ball on the wall; stepping up and down on a ladder); and $N_4$ (crouching/lifting throwing a ball on the wall). In the stepping up and down exercise on the ladder, the patient goes up and down only one step 30 cm high. A ball

approximately 70 cm in diameter and 200 g will be used for ball exercises. Before attempting protocol exercises, patients will be instructed on the proper way to accomplish each exercise. Demonstration and realization of up to three slow repetitions of each exercise will be performed. Training will start and stop according to the safety criteria of the expert consensus and the recommendations for active mobilization [24]. The safety criteria are divided into four categories (respiratory, cardiovascular, neurological and others) using a standard color system [24]: red indicates the need for caution, with the risk of an adverse event; yellow indicates that the application of the exercise protocol is possible, but only after further analysis with the ICU multidisciplinary team; and green indicates that the patient is safe to apply the exercise protocol. Changes in conditions and in the direction of trends in respiratory, cardiovascular and neurological variables outside the safety intervals [24] imply the immediate suspension of exercises. If normalization of the altered physiological parameter occurs, within 5 minutes, the exercise protocol is reestablished and monitoring continues. If there is no normalization, the team will be informed and the attempt to apply the protocol will occur after 24 hours.

### Intervention outcomes

Socio-demographic, health, and lifestyle data will be collected from the patients through the report of the legal guardians and when possible, the patients themselves. Data of the medical aspects of the study will be collected through the patient's chart, ICU monitors, and direct communication with the multidisciplinary team. The data will be recorded in an appropriate form by properly trained researchers. Researchers who collected study data will not be part of the hospital staff.

All outcomes will be assessed according to the outcome assessment recommendations of Young et al. [25]. The primary outcome will be the length of stay in the ICU. The secondary outcome measures included:

- Cross-sectional area and muscle thickness of the rectus femoris and biceps brachii measured by ultrasound, using an 8-MHz 5.6-cm linear transducer array (PLM805, Toshiba Medical Systems, Crawley, UK) immediately after randomization and at intervals of 48 h for 28 days or till discharge from the ICU, whichever occurred earlier. For evaluation of the rectus femoris, the transducer will be placed perpendicular to the long axis of the thigh on its superior aspect, three-fifths of the distance from the anterior superior iliac spine to the superior patellar border [26]. Biceps brachii will be evaluated at the midline between origin and insertion [27]. The transducer will be positioned with slight pressure.

- Change in muscle strength will be evaluated immediately after randomization, after 3 days and at intervals of 7 days, up to the 28th day or till discharge from the ICU. Muscle strength will be measured using the following: (1) Medical Research Council sum-score (MRC) [28]; (2) handgrip, using a hydraulic dynamometer (Saehan Corporation SH5001, Korea), with the handle pressed for at least 6 seconds in three trials with a 1-min intervals; the highest value of manual grip strength in kilograms (Kgf) will be considered for the analysis [29]; (3) hand-held dynamometer of elbow flexion and knee extension (model 01163; Lafayette Instrument, Ind) [30]; and (4) five Sit to Stand Tests [31].

- Functional status at 28 days after randomization or discharge from ICU will be measured using the FSS [23], Surgical Optimal Mobilization Score (SOMS) [12], and timed up-and-go score [32].

- ICUAW will be determined by a score lower than 48 on the MRC sum-score, according to Vanpee et al. [28].

- Days with mechanical ventilation and mortality at 28 days after randomization or discharge from the ICU.

## Sample size calculations

The sample size calculation will be based on a mean difference in the length of hospital stay of 2.5 days; considering a confidence interval of 95%, power of 80%, and a ratio of 1: 1 between the two groups. Based on the averages of the study by Dong et al. [33] (12.7±4.1 vs 15.2±4.5 days), a sample size of 94 patients was obtained, 47 for each group. After adjusting 20% for missing, death, loss to follow-up, and withdrawals, the corrected sample was 118 subjects. The study by Dong et al. [33] was chosen because it had a population similar to that in the present study. Among other aspects of population similarity, the study involved patients able to perform activities independently and with a probability of <50% for death at 6 months. The sample size calculation will be performed using the module that calculates sample sizes to compare two averages available on the website: https://www.openepi.com/SampleSize/SSMean.htm.

This sample also will be sufficient to contemplate the functional independence objective, based on the percentage of patients returning to their functional state at the time of hospital discharge (FSS = 35). A final sample size of 86 patients was obtained, 43 for each group, based on 50% of patients returning to the previous functional state at the time of hospital discharge [34] and assuming that 80% of the patients remained in the intervention group at the time of hospital discharge.

## Data analysis

For the analysis of demographic and clinical data, descriptive statistics will be used. The data of continuous variables will be analyzed with measures of central tendency and dispersion. The categorical variable data will be analyzed with frequency measurements. To perform the inferential statistics, tests for normality and homogeneity of variance will be performed. Since the data will be normally distributed, analysis of variance (ANOVA) will be used to compare the mean differences of the variables, followed by the Bonferroni post hoc test. If there will be no normal distribution, the Kruskal-Wallis test will be used, followed by the Mann-Whitney post-hoc test. $\chi 2$ tests will be used to evaluate the differences in the categorical variables between the groups.

Kaplan-Meier analysis will be used to estimate ICU LOS, mortality, and mechanical ventilation time, and a log-rank will be used to assess the difference between the groups. Poisson or negative binomial model will be used to estimate the incidence rate ratio. Stratified analyses will be performed according to age, initial functional status, and the Acute Physiology and Chronic Health Enquiry (APACHE-II) score to identify potential confounders. Cohen's $d$ effect sizes will be also calculated. Before carrying out the analyzes, test methods assumptions will be verified.

Mean and median differences and 95% confidence intervals will calculated. The significance threshold will be used $P < .05$ for each outcome and testing will be used two-sided. The statistical software used will be used IBM SPSS for Windows 21.0. All study analyses will be used performed according to the intention to treat.

## Trial organization, monitoring, and safety

The research team will include the authors listed in this protocol. The principal investigator (R.S.Q) will manage the data flow and will carry out audits of procedures, registration and intervention adherence throughout the process of this study. The associate investigators listed

in this protocol (M.G.N, B.P.M, and M.B.S), will monitor the data-collection process and data integrity and perform periodic evaluations during the course of the data-collection phase.

The overall risk level involved in the area studies is low, usually less than 4% [24], and involved transient reductions in saturation, hypotension, or hypertension, which are normalized at rest [24]. One researcher (R.S.Q) will monitor for adverse effects on mobilization.

Data collection and registration will be performed with the support of an electronic form developed specifically for the study through Google forms. The data entered in the form will be automatically recorded in a restricted access spreadsheet. The database will be sequentially monitored by researchers who will be not directly part of the collection.

### Ethics

The study was approved by the institutional committee of Universitary Hospital Professor Edgard Santos of the Federal University of Bahia (UFBA), Salvador, Bahia, Brazil (number: 2.371.933). Informed consent will be obtained from potential trial participants or authorized surrogates (family member or guardian).

## Discussion

Current evidence shows that critically ill patients who remain in bed rest develop neuromuscular disorders. Early and progressive active mobilization of these patients has the potential to reduce the length of hospital stay and improve functionality. However, no clinical trial has completely controlled the dose of mobilization to date.

### Potential impact and significance of the study

The PROMOB protocol is the first protocol of progressive mobility [5] developed for critical patients that presents all the training load variables described according to the recommendations of the TIDieR checklist [35], CONSORT guidelines for non-pharmacological interventions [36], and Consensus on Exercise Reporting Template (CERT) [37]. PROMOB is a fast, objective protocol and is easy to replicate. The exercise set and the training dose control were elaborated from a systematic review [5] controlling important exercise variables, such as intensity, number of repetitions, execution time, rest time, and objective progression criteria, which were not controlled in other studies in the field [5].

An important difference in this protocol is its ability to explore the capability of performing a controlled intensity of exercise in the ICU in an individualized manner, feasible with safety parameters, specific instruments for measuring the functional outcomes, and focus on interventions for transfer and locomotion activities. The present clinical trial protocol has specific screening criteria to recognize patients with minimal prior functional capacity (Barthel index), collaborative capacity (ability to interact with the physiotherapist), and capability of performing active movement and transfers without the need for equipment or excessive staff overload, ensuring that the patients can perform the tasks (FSS-ICU ≥3). The inclusion of these screening elements selects a specific population of critical patients who may present a "healthy trajectory" after ICU stay if preventable comorbidities were avoided [38], as with the case involving ICUAW.

Another potential impact of the study is that we evaluated a mobilization intervention of greater intensity at an early stage. A recent meta-analysis [3] stated that despite the difficulty in conceptualizing the dose of mobilization due to poor description of the training variables in the studies involved, early and low-dose rehabilitation had benefits in terms of the number of days living and out of the hospital within 180 days, unlike late and higher dose rehabilitation, which did not achieve these results. The dose of mobilization may affect the outcomes of

critical patients; therefore, the description and integral control of the mobilization dose in clinical trials is essential.

Other well-designed progressive active mobilization protocols have been published [39]; however, these were pilot studies with different mobility training characteristics. A recent clinical trial [40] evaluated the impacts of a progressive mobility program on critically ill patients, however, the exercise protocol presented, among others, differs from the present protocol in that it uses equipment (cycle ergometer, electrical stimulation, video game, among others). In addition, the study [40] does not describe the dosage of trunk and walking exercises. Aspects related to the intensity of the exercises are also not mentioned [40]. The PROMOB emphasizes progressive interval training, with run time control, rest, and intensity.

## Strengths and weaknesses of the study

One of the strengths of the study is the study design, which involved blinding of the patients and functional assessors, presenting an adequate sample for the proposed objectives as well as an adequate description of the interventions offered in both groups. Recently, a Cochrane review [4] described that there is insufficient evidence of good quality to support the benefits of mobilization for improved physical function or performance, muscle strength, and quality of life. The present protocol, based on an individualized prescription and focusing on transference and locomotion, as well as on the monitoring of effort intensity (using the Borg scale), may expose patients to an adequate dose so that the effects of exercise can promote the necessary changes in the cardiorespiratory system and skeletal muscle. Another important point is that the protocol will have an early start and will be based on active mobilization, which will reduce some confounding variables observed when this mobilization is initiated at a later stage.

The protocol has some limitations, such as the fact that it will be performed initially in only one center; furthermore, the ICU had a mixed profile of patients, including clinical and surgical patients. Another confounding variable will be the absence of a prior functional status control for physical exercise.

## Supporting information

**S1 Checklist. CONSORT 2010 checklist of information to include when reporting a randomised trial**[*].
(DOCX)

**S1 File.**
(DOCX)

**S2 File.**
(DOCX)

**S3 File.**
(DOCX)

## Author Contributions

**Conceptualization:** Rodrigo Santos de Queiroz, Micheli Bernardone Saquetto, Bruno Prata Martinez, Mansueto Gomes-Neto.

**Formal analysis:** Mansueto Gomes-Neto.

**Investigation:** Rodrigo Santos de Queiroz, Micheli Bernardone Saquetto, Mansueto Gomes-Neto.

**Methodology:** Rodrigo Santos de Queiroz, Micheli Bernardone Saquetto, Bruno Prata Martinez, Mansueto Gomes-Neto.

**Project administration:** Rodrigo Santos de Queiroz, Micheli Bernardone Saquetto, Bruno Prata Martinez, Bianca Bigogno Reis Cazeta, Mansueto Gomes-Neto.

**Supervision:** Rodrigo Santos de Queiroz, Bruno Prata Martinez, Bianca Bigogno Reis Cazeta, Mansueto Gomes-Neto.

**Validation:** Mansueto Gomes-Neto.

**Visualization:** Carol Hodgson.

**Writing – original draft:** Rodrigo Santos de Queiroz, Mansueto Gomes-Neto.

**Writing – review & editing:** Rodrigo Santos de Queiroz, Carol Hodgson, Mansueto Gomes-Neto.

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
