## [Decision Letter · Decision Letter 0]

25 Jun 2020

PONE-D-20-03375

Progressive active mobilization with dose control and training load in critically ill patients (PROMOB): protocol for a randomized controlled trial

PLOS ONE

Dear Dr. Queiroz,

Thank you for submitting your manuscript to PLOS ONE. After careful consideration, we feel that it has merit but does not fully meet PLOS ONE’s publication criteria as it currently stands. Therefore, we invite you to submit a revised version of the manuscript that addresses the points raised during the review process. Please note that the methods, mainly in terms of the intervention details to explain the progressive nature of the exercise protocol, and detailed methods that allow reproducibility. In addition, a stronger rationale should clarify why strict mobilization protocols are needed in ICU given the high variability of possible conditions in such setting. Also, a proper contextualization with previous but also current literature is needed, as the reviewers noted in the comments and recommendations the authors may see below in this email.

We look forward to receiving your revised manuscript.

Kind regards,

Jose María Blasco, Ph.D.

Academic Editor

PLOS ONE

Reviewers' comments:

Reviewer's Responses to Questions

**Comments to the Author**

1. Does the manuscript provide a valid rationale for the proposed study, with clearly identified and justified research questions?

Reviewer #1: Yes

Reviewer #2: Yes

Reviewer #3: Partly

2. Is the protocol technically sound and planned in a manner that will lead to a meaningful outcome and allow testing the stated hypotheses?

Reviewer #1: Yes

Reviewer #2: Yes

Reviewer #3: Partly

3. Is the methodology feasible and described in sufficient detail to allow the work to be replicable?

Reviewer #1: Yes

Reviewer #2: Yes

Reviewer #3: No

4. Have the authors described where all data underlying the findings will be made available when the study is complete?

Reviewer #1: No

Reviewer #2: Yes

Reviewer #3: Yes

5. Is the manuscript presented in an intelligible fashion and written in standard English?

Reviewer #1: Yes

Reviewer #2: Yes

Reviewer #3: Yes

6. Review Comments to the Author

You may also provide optional suggestions and comments to authors that they might find helpful in planning their study.

Reviewer #1: I will focus on methods and reporting. Level of English needs to be improved but it is not bad enough to prevent publication. The abstract is informative and balanced. Randomisation is appropriate. The description of the standard treatment and the intervention is clear.

I was confused by the description of the primary outcome: "The primary outcome will be ICU stay at 28 days after randomization or discharge from the ICU, whichever occurred earlier". If I understand this correctly, the outcome is binary, status at +28 days (discharged or not). The way it is phrased I find confusing.

There are perhaps too many secondary outcomes for such a small sample, and there is a risk that something will be statistically significant by chance. So i'd urge the researchers not to make too much of the analyses of secondary outcomes and to focus on effect sizes rather than p-values.

Power calculations are OK but hospital stay is not normally distributed. So using a normal approximation for the power calculation has its issues (e.g. see https://www.sciencedirect.com/science/article/pii/S1098301517301298). Not that it's the end of the world so I'll leave to the authors to revisit if they feel they need to. Otherwise acknowledge as a limitation (the normality assumptions). Also doesn't really match the outcome, which is binary(?), unless i completely misunderstood it. Finally, I can't see the baseline level there which is needed to put into context the size of the hypothesised effect of the intervention. If on average stay is 10 days, then this is a massive effect, for example.

Analysis plan is appropriate, but make sure methods assumptions are met, e.g. proportional hazards for Cox. Again, I'm not clear exactly what the primary outcome is and hence what is the relevant analysis. If the outcome is LOS, why not use a Poisson or negative binomial model, for example.

Another concern is that the sample is relatively small. Are the authors confident balance will be achieved on all covariates of interest or a more deterministic matching approach is needed?

Reviewer #2: The topic related to the presented protocol is relevant but not new. Authors describe their single-centre trial design to evaluate the effectiveness of progressive early mobilization process in ICU.

Recently another group from Sao Paulo-Brazil have published similar data on a progressive protocol applied to ICU-patients (see Stripari et al. Crit Care Med 2020; 48:491–497)

Authors of the present submission should be therefore quote this last evidence from literature and very well explained which is the substantial difference between protocols and, moreover, what is expected from their trial as compared with the already published evidences and results.

Reviewer #3: I thank the authors for the opportunity to review this trial protocol. The authors plan to test a progressive mobilization intervention for critically ill patients in ICU. There are several strengths of this protocol, including:

-prospectively registered

-adheres to the SPIRIT statement

-important endpoints (LOS, ICUAW, Days with mechanical ventilation, Mortality)

-plan for allocation concealment and intention to treat

-plan for blinding of assessors

However, despite these strengths, I have several concerns about the rationale and intervention that need to be addressed.

Finding the 'optimal training scheme' for ICU patients seems to be a key rationale for this trial. I find this problematic. I imagine trying to find the optimal loading protocol for patients in ICU would be extremely difficult given the range of conditions/presentations. In this setting, wouldn't tailoring mobilisation to a patients level of function be more important than trying to find the optimal dose for everyone? This might be the reason guidelines don't specify an exact loading protocol for critically ill patients in ICU.

After reading the methods, it seems the progressive active mobilisation intervention will be 'added' to usual care, which means participants in the intervention will receive a greater volume of treatment/mobilisation than participants in the usual care group. If the intervention is effective, how will the researchers determine that the improvement was due to the protocol or simply due to additional treatment time?

If I have misunderstood the protocol, how will you control for treatment time between groups?

Looking at Figure 1, I am unsure how the intervention would be sufficiently different to usual care. Wouldn't all therapists gradually increase the intensity/volume of mobilisation for patients in ICU? Whether they do it intentionally, therapists will usually start by prescribing bed exercises and gradually build up to walking and then walking up stairs. I am concerned there wont be enough contrast between the groups to show something meaningful. Again, apologies if I have misunderstood the protocol.

Other comments:

LOS is missing from aim at the end of the introduction

I am unsure biceps brachii cross-sectional area would be the most important upper limb muscle to assess. I imagine loading through the arms (e.g. during sit to stand or during mobilisation in a frame) would cause more adaptations in the triceps than biceps (due to the extension moment arm of these activities)

7. PLOS authors have the option to publish the peer review history of their article (what does this mean?). If published, this will include your full peer review and any attached files.

Reviewer #1: No

Reviewer #2: No

Reviewer #3: Yes: Josh Zadro

---

## [Author Response · Author response to Decision Letter 0]

17 Jul 2020

Response to reviewers

Dear editor and reviewers,

I am immensely grateful for all contributions to this manuscript. Below are the responses to the reviewers.

Regards,

Editor

• As requested and in accordance with the recommendations presented, a revised version of the manuscript follows with all points raised during the review process.

• The changes were made in the body of the text, and other details were explained in the responses to the reviewers.

• According to the reviewers' comments, the particularities of this protocol and its applicability were explained and a contextualization was made with the most current literature.

• Any doubts, or even need for further adjustments, we are available.

• The current ethics statement has been changed to include the full name of the ethics committee (Universitary Hospital Professor Edgard Santos of the Federal University of Bahia - UFBA) - (Page 11; Line 275-277)

• The requested text was added to the registration platform ("All relevant data from this study will be made available upon study completion.")

Reviewer #1:

Questioning: “I was confused by the description of the primary outcome: "The primary outcome will be ICU stay at 28 days after randomization or discharge from the ICU, whichever occurred earlier". If I understand this correctly, the outcome is binary, status at +28 days (discharged or not). The way it is phrased I find confusing.”

• Answer: Thank you for your observation. The description of the primary outcome has been improved (Page 7; Line 189)

Questioning: “There are perhaps too many secondary outcomes for such a small sample, and there is a risk that something will be statistically significant by chance. So i'd urge the researchers not to make too much of the analyses of secondary outcomes and to focus on effect sizes rather than p-values.”

• Answer: Thank you for your observation. In this protocol proposal, we put more secondary outcomes to perform an exploratory analysis only. We appreciate the observation and we will pay attention to the effect sizes.

 Questioning: “Power calculations are OK but hospital stay is not normally distributed. So using a normal approximation for the power calculation has its issues (e.g. see https://www.sciencedirect.com/science/article/pii/S1098301517301298). Not that it's the end of the world so I'll leave to the authors to revisit if they feel they need to. Otherwise acknowledge as a limitation (the normality assumptions). Also doesn't really match the outcome, which is binary(?), unless i completely misunderstood it. Finally, I can't see the baseline level there which is needed to put into context the size of the hypothesised effect of the intervention. If on average stay is 10 days, then this is a massive effect, for example. Analysis plan is appropriate, but make sure methods assumptions are met, e.g. proportional hazards for Cox. Again, I'm not clear exactly what the primary outcome is and hence what is the relevant analysis. If the outcome is LOS, why not use a Poisson or negative binomial model, for example.”

• Answer: Thanks for your observation, suggestion accepted. The length of stay in the ICU will be assessed as a continuous variable, we improved the text of the primary outcome to make this clearer. We read the recommended article on statistical analysis and we integrated Poisson or negative binomial in the analysis plan. We also added that before carrying out the analyzes, test methods assumptions will be verified. (Page 10; Line 248-249 and 252). 

Questioning: “Another concern is that the sample is relatively small. Are the authors confident balance will be achieved on all covariates of interest or a more deterministic matching approach is needed?”

Answer: Thank you very much for this careful observation. We are confident that there will be a balance in the distribution between the covariates of interest. We did the sample size calculation and the randomization process tends to balance the variables. If any variable is not balanced, it will be used to control the analysis. In other clinical trials in the area, which had a similar sample, there was a balance between the main covariates.

Reviewer #2:

Questioning: “The topic related to the presented protocol is relevant but not new. Authors describe their single-centre trial design to evaluate the effectiveness of progressive early mobilization process in ICU. Recently another group from Sao Paulo-Brazil have published similar data on a progressive protocol applied to ICU-patients (see Stripari et al. Crit Care Med 2020; 48:491–497). Authors of the present submission should be therefore quote this last evidence from literature and very well explained which is the substantial difference between protocols and, moreover, what is expected from their trial as compared with the already published evidences and results”

• Answer: Thanks for your observation, suggestion accepted. We added the reference of the article by Schujmann et al (2020) as recommended, mentioning the differences between the protocols in the discussion. The exercise protocol presented by Schujmann et al (2020), among others, differs from the present protocol in that it uses equipment (cycle ergometer, electrical stimulation, video game, among other equipment). In addition, the study does not describe the dosage of trunk and walking exercises. Aspects related to the intensity of the exercises are also not mentioned. The PROMOB emphasizes progressive interval training, with run time control, rest, and intensity. By describing the exercise dosage more appropriately and because it does not require equipment, the PROMOB protocol can be more easily replicated in clinical practice. (Page 12-13; Line 321-326)

SCHUJMANN, Debora Stripari et al. Impact of a Progressive Mobility Program on the Functional Status, Respiratory, and Muscular Systems of ICU Patients: A Randomized and Controlled Trial. Critical care medicine, v. 48, n. 4, p. 491-497, 2020.

Reviewer #3:

Questioning: “Finding the 'optimal training scheme' for ICU patients seems to be a key rationale for this trial. I find this problematic. I imagine trying to find the optimal loading protocol for patients in ICU would be extremely difficult given the range of conditions/presentations. In this setting, wouldn't tailoring mobilisation to a patients level of function be more important than trying to find the optimal dose for everyone? This might be the reason guidelines don't specify an exact loading protocol for critically ill patients in ICU.”

• Answer: Thanks for your observation. Indeed, finding an ideal exercise protocol for critically ill patients has been a great challenge for researchers around the world. In addition to being one of the first clinical trial protocols that adequately describes all exercise prescription variables for critically ill patients, PROMOB stratifies the exercises by patients' functional level. This adaptation was made based on the FSS-ICU (Functional Status Score for the ICU). This has been a concern of ours since the project was conceived. In addition to adapting to the functional level, we realized that even within a functional level, some patients have different exercise capacity. In this sense, there is exercise variation also according to the performance presented (Adequate performance = Able to perform 8 series of each movement, alternating 20 seconds of execution with 10 seconds of rest, without discomfort - Borg <6; scale 0-10) . We believe that this rationale makes the prescription more individualized. The main guidelines for the prescription of exercises in the ICU are not yet able to establish an ideal treatment protocol due to the complexity that involves the care of critically ill patients, but also due to an incomplete description of the protocols presented in studies in the area. As presented in the manuscript, it is strongly encouraged that studies adequately describe the exercise dose employed. Due to the ethical aspects involved, that is, it is currently not possible to have a control group that does not receive mobilization, the implementation of new protocols added to the usual care is the most widely used strategy. Thus, patients in the intervention group receive a greater volume of training than the control. Once the intervention proves to be effective, the need to implement the usual care is evident.

Questioning: “After reading the methods, it seems the progressive active mobilisation intervention will be 'added' to usual care, which means participants in the intervention will receive a greater volume of treatment/mobilisation than participants in the usual care group. If the intervention is effective, how will the researchers determine that the improvement was due to the protocol or simply due to additional treatment time?

If I have misunderstood the protocol, how will you control for treatment time between groups? Looking at Figure 1, I am unsure how the intervention would be sufficiently different to usual care. Wouldn't all therapists gradually increase the intensity/volume of mobilisation for patients in ICU? Whether they do it intentionally, therapists will usually start by prescribing bed exercises and gradually build up to walking and then walking up stairs. I am concerned there wont be enough contrast between the groups to show something meaningful. Again, apologies if I have misunderstood the protocol.”

• Answer: Thanks for your observation. Both groups will receive the same standard of care as usual. The treatment time itself is not the main training load variable, we opted for a protocol that involves a series of exercises with a short rest time and with control and intensity increase. The intervention group has a longer time to perform daily exercises because they have to perform the study protocol. The execution time of the protocol exercises is variable and depends on the patient's performance. The time to implement the protocol is short, as we opted for a practical and feasible protocol to be implemented in clinical practice (8 series of each movement, alternating 20 seconds of execution with 10 seconds of rest).The increase in the volume of training within the usual care will occur both in the control group and in the intervention group, it will be in accordance with criteria established by the ICU team, we researchers will not have any influence. The contrast between the groups will occur precisely because of the addition of our protocol, which is systematic, progressive and with training load control. With the training load control, not only is the volume of exercise systematically increased in the intervention group, but also the intensity. The proposal is to add this protocol to the usual care, and not to compare the protocol with the usual care. Obtaining an effective result, we will point to the need to add this protocol to the usual care to improve outcomes in the ICU.

Questioning: “LOS is missing from aim at the end of the introduction”

• Answer: Thanks for your observation, suggestion accepted. LOS has been added (Page 3; Line 54)

Questioning: “I am unsure biceps brachii cross-sectional area would be the most important upper limb muscle to assess. I imagine loading through the arms (e.g. during sit to stand or during mobilisation in a frame) would cause more adaptations in the triceps than biceps (due to the extension moment arm of these activities)”

• Answer: Thanks for your observation. We conducted a systematic review on the use of muscle ultrasound in critically ill patients and the cross-sectional area of the biceps brachii is the muscle of choice for inferring the muscle mass of the upper limbs. We agree with your observation, however we did not find studies involving critical patients who evaluated the triceps. In order to follow evaluation parameters already described in the literature and in discussion with the radiology team that is part of our project, we chose to evaluate the biceps brachii.

---

## [Decision Letter · Decision Letter 1]

17 Aug 2020

Progressive active mobilization with dose control and training load in critically ill patients (PROMOB): protocol for a randomized controlled trial

PONE-D-20-03375R1

Dear Dr. Queiroz,

We’re pleased to inform you that your manuscript has been judged scientifically suitable for publication and will be formally accepted for publication once it meets all outstanding technical requirements.

Kind regards,

Jose María Blasco, Ph.D.

Academic Editor

PLOS ONE

Additional Editor Comments (optional):

Reviewers' comments:

Reviewer's Responses to Questions

**Comments to the Author**

1. Does the manuscript provide a valid rationale for the proposed study, with clearly identified and justified research questions?

Reviewer #1: Yes

Reviewer #2: Yes

Reviewer #3: Yes

2. Is the protocol technically sound and planned in a manner that will lead to a meaningful outcome and allow testing the stated hypotheses?

Reviewer #1: Yes

Reviewer #2: Yes

Reviewer #3: Yes

3. Is the methodology feasible and described in sufficient detail to allow the work to be replicable?

Reviewer #1: Yes

Reviewer #2: Yes

Reviewer #3: Yes

4. Have the authors described where all data underlying the findings will be made available when the study is complete?

Reviewer #1: Yes

Reviewer #2: Yes

Reviewer #3: Yes

5. Is the manuscript presented in an intelligible fashion and written in standard English?

Reviewer #1: Yes

Reviewer #2: No

Reviewer #3: Yes

6. Review Comments to the Author

You may also provide optional suggestions and comments to authors that they might find helpful in planning their study.

Reviewer #1: I am happy with the changes the authors have made and their responses. Perhaps more of what was discussed could have made it into the paper, but I'll leave to the authors to decide that. As long as they pay attention to the methodological issues raised, it will be fine.

Reviewer #2: Authors have satisfactorily responded to my query and concern, also they updated refs as suggested in their revised version.

Reviewer #3: I am satisfied with the authors response to my concerns. I was initially concerned about the originality of this piece of work and whether there was enough contrast between the intervention and control group. The authors have highlighted the need for trials to adequately report their exercise protocols and the challenge of testing exercise interventions when usual care involves exercise. Their response highlights an in-depth understanding of their field and the need to test clearly defined protocols. I am happy for this protocol to be published and wish the authors the best of luck with their trial.

7. PLOS authors have the option to publish the peer review history of their article (what does this mean?). If published, this will include your full peer review and any attached files.

Reviewer #1: No

Reviewer #2: No

Reviewer #3: **Yes: **Joshua Zadro

---

## [Editor Report · Acceptance letter]

25 Aug 2020

PONE-D-20-03375R1 

Progressive active mobilization with dose control and training load in critically ill patients (PROMOB): protocol for a randomized controlled trial 

Dear Dr. Queiroz:

I'm pleased to inform you that your manuscript has been deemed suitable for publication in PLOS ONE. Congratulations! Your manuscript is now with our production department. 

Kind regards, 

on behalf of

Dr. Jose María Blasco 

Academic Editor

PLOS ONE